# Assessment of Thermochromic Packaging Prints’ Resistance to UV Radiation and Various Chemical Agents

**DOI:** 10.3390/polym15051208

**Published:** 2023-02-27

**Authors:** Sonja Jamnicki Hanzer, Rahela Kulčar, Marina Vukoje, Ana Marošević Dolovski

**Affiliations:** Faculty of Graphic Arts, University of Zagreb, 10000 Zagreb, Croatia

**Keywords:** intelligent packaging, thermochromic inks, chemical resistance, UV radiation, color degradation

## Abstract

Thermochromic inks, also known as color changing inks, are becoming increasingly important for various applications that range from smart packaging, product labels, security printing, and anti-counterfeit inks to applications such as temperature-sensitive plastics and inks printed onto ceramic mugs, promotional items, and toys. These inks are also gaining more attention as part of textile decorations and can also be found in some artistic works obtained with thermochromic paints, due to their ability to change color when exposed to heat. Thermochromic inks, however, are known to be sensitive materials to the influence of UV radiation, heat fluctuations, and various chemical agents. Given the fact that prints can be found in different environmental conditions during their lifetime, in this work, thermochromic prints were exposed to the action of UV radiation and the influence of different chemical agents in order to simulate different environmental parameters. Hence, two thermochromic inks with different activation temperatures (one being cold and the other being body-heat activated), printed on two food packaging label papers that differ in their surface properties were chosen to be tested. Assessment of their resistance to specific chemical agents was performed according to the procedure described in the ISO 2836:2021 standard. Moreover, the prints were exposed to artificial aging to determine their durability when exposed to UV radiation. All tested thermochromic prints showed low resistance to liquid chemical agents as the color difference values were unacceptable in all cases. It was observed that the stability of thermochromic prints to different chemicals decreases with decreasing solvent polarity. Based on the results obtained after UV radiation, its influence in terms of color degradation is visible on both tested paper substrates, but more significant degradation was observed on the ultra-smooth label paper.

## 1. Introduction

Today, thermochromic printing inks are used in various fields from the food industry to industrial design. They are printed on the packaging or applied on a label so that the consumer can determine by the color of the ink whether the product is ready for consumption. Active and intelligent packaging has become very popular and in demand in the modern packaging industry. They enable greater added-value, increased security, and additional opportunities for marketing promotion [1,2,3,4]. Moreover, thermochromic inks are increasingly interesting to many artists who use these types of inks in their creative expressions [5,6,7,8,9,10,11,12,13]. Thermochromic inks are part of a group of chromogenic inks that have the characteristic of changing their color due to some external influence. Leuco-dye-based thermochromic (TC) inks change their color due to temperature changes. One of the most important elements of smart packaging is a temperature sensitivity indicator that shows the current temperature of the product. These TC indicators are usually simple in design, and with their dynamic color change, add a functional role to the packaging.

TC color change can go from colorful to colorless, colorless to colorful, or they can change from one color to another. Also, TC colorants can be successfully mixed with conventional inks to obtain a wider range of colors [14,15,16,17,18,19,20]. This can expand creative possibilities for designers and artists on varied materials and applications. TC dynamic color change can be reversible or irreversible. The reversible color change can last for 10s of years if the print is protected from direct sunlight, extremely hot temperatures, and aggressive solvents. In recent years, more research has been carried out on TC inks to determine the factors that negatively affect the TC color change effect and to find solutions to preserve it if possible. Each TC ink has its own fixed activation temperature (T_A_) at which the color change starts, and today thermochromic inks can be obtained at many different activation temperatures [21]. TC inks based on leuco dyes generally consist of an encapsulated three-part system including a dye that changes color, a color developer, and a solvent [22]. At temperatures below the T_A_, the solvent is in a solid state enabling the dye and color developer to form a color which results in a full color effect. When the temperature reaches the T_A,_ the solvent becomes liquid, keeping color developer and the leuco dye apart making the ink appear transparent or translucent [23,24,25,26,27]. The composition of TC inks includes a mixture of TC microcapsules dispersed in the ink’s vehicle [24,28,29,30]. The main disadvantage of this microencapsulated TC system is its poor stability to some external influences [31,32]. Reversible discoloration can result in ruptures of microcapsules containing dye [33]. How to protect this system, which is responsible for the dynamics of color change, is a broader subject of our research. Some examples include choosing a suitable printing substrate or the use of nanomodified coatings [34,35,36]. 

In order to preserve the dynamics of color change and functionality of thermochromic inks as indicators for packaging applications, this paper will investigate the factors that could negatively affect their durability during their life cycle with the aim of maintaining the functional role of TC inks as an additional value on packaging. Thermochromic prints on packaging (labels) could be exposed to the effect of different chemical agents (ethanol in alcoholic products, citric acid in juices, vegetable oils, water, etc.…) during the use of the products on which they are applied as indicators, which can affect their functionality. In our previous research, we found that exposure of TC prints to specific liquid agents, such as citric acid and ethanol caused severe damage to the prints which also led to the bleeding of the colorants from the prints [37]. In addition to different chemical agents, the packaging can be exposed to different storage and transportation conditions. Moreover, during their shelf life, packaging can be exposed to UV radiation from light sources in the supermarket as well. All these conditions can negatively affect the packaging properties, in this case the properties of thermochromic indicators [38]. A variety of different complex chemical reactions occur during the natural or artificial degradation of paper/cardboard and prints due to different environmental and storage conditions, especially UV radiation, temperature, and humidity, which can lead to different deteriorations of paper-based packaging materials. In the last 10 years, several studies have been published that dealt with low light fastness issues of TC prints [31,33,39,40,41], but there is still a lack of research related to the resistance of TC prints to various chemical agents as evidenced by the relatively small number of available scientific papers on the subject. 

This research aims to analyze the resistance of UV-cured thermochromic prints to specific liquid chemical agents and UV radiation. For that reason, the prints obtained on two different paper substrates were subjected to artificial aging and were also treated with three liquid agents to prove the prints’ resistance to liquids and chemicals. Chemical resistance of prints was observed based on the optical deterioration of prints after treatments with liquid agents compared against untreated prints and by determination of bleeding of the colorants into the receptor surface. Prints’ resistance to UV radiation was evaluated by observing the discoloration and any change in the prints.

## 2. Materials and Methods

### 2.1. Printing Substrates

As printing substrates, two food packaging label papers of similar basis weight and ISO brightness (R_457_) values were selected with properties given by the manufacturer, presented in Table 1. 

The Niklakett Premium Fashion—NPF (Brigl & Bergmeister GmbH, Niklasdorf, Austria) substrate was a high gloss embossed label paper with a textured surface and about four times lower smoothness compared to a Chromolux 700—CHR (Zanders Paper GmbH, Bergisch Gladbach, Germany) paper characterized by ultra-smooth high gloss surface. Both papers were suitable for all common printing processes. Moreover, Chromolux 700 was approved for direct food contact. Also, both papers had a functional coating on the reverse side as they were designed to be used as labels for beverages.

### 2.2. Printing Inks

Two thermochromic commercially available UV screen inks were chosen for printing. These inks were colored below a specific temperature and changed to another color as they were heated. One ink was colored in orange below its activation temperature (T_A_ = 12 °C) and changed to yellow above it (hereinafter OY-12). Due to the low activation temperature, this ink is also called low temperature (or cold activated ink) commonly used for print media to be chilled in the refrigerator or for other cold reactions. Application examples include cold-temperature indicators, cold packaging, and other low temperature applications. Another ink was colored in purple below its activation temperature (T_A_ = 31 °C) and changed to pink above it (hereinafter PP-31). This ink was the type of body-heat activated ink that is designed to show color at normal room temperature and to change color when warmed up. This can be done by rubbing a finger or breathing warm air on the application. Application examples of this type of ink include media packaging, direct mail pieces, stickers, and labels where a unique interactive consumer experience is desired. Both inks were reversible inks meaning that the original color was restored upon cooling (Figure 1 and Figure 2). These inks were made by mixing two types of colorants: thermochromic leuco dyes encapsulated inside the microcapsule (TC microcapsules) and conventional pigments [30,42]. We can assume that orange (OY-12) ink was made by the addition of red leuco dyes to conventional yellow pigments, and purple (PP-31) ink was made by a combination of conventional pink pigments with blue leuco dyes. 

The manufacturer states that these inks are sensitive to adverse environmental conditions. Inks should not be exposed to UV and some fluorescent lights for a long time, nor to direct sunlight for more than a few days, as this can degrade color intensity and color-changing characteristics of the ink. The manufacturer does not recommend exposing these inks to elevated temperatures for extended periods of time, as prolonged exposure to 38 °C (100 °F) or higher, can degrade the color change and intensity of the product. In addition, these inks are sensitive to certain chemicals, that is, wet ink should not come into contact with any solvents. 

### 2.3. Printing Trials

Printing was conducted in laboratory conditions by a semi-automatic screen-printing device Siebdruckgeräte von Holzschuher KG., Wuppertal, using a mesh of 62–64 lin/cm. The prints were printed in full tone. After printing, prints were cured using a Technigraf Aktiprint L 10-1 UV dryer (30 W/cm). The characteristics of inks given by the producer are presented in Table 2. 

### 2.4. Assessment of Prints’ Resistance to Specific Chemical Agents

Assessment of prints’ resistance to specific chemical agents was done following the standard method ISO 2836:2021 [43]. International Standard ISO 2836:2021 in the field of the printing industry defines methods of assessing the resistance of prints to liquid and solid agents, solvents, varnishes, and acids [43]. For this study, thermochromic prints were exposed to distilled water, citric acid, and ethanol, which were chosen as test agents to simulate the exposure of smart beverage labels to water, alcohol, and juice of citrus fruits. For test procedures where water and citric acid were used as liquid agents, the prints were cut to dimensions of approx. 2 cm × 5 cm. For the determination of the resistance to citric acid, the printed test piece was brought into contact with filter papers previously saturated with 5% citric acid and placed under a 1 kg load for 1 h at ambient temperature (23 ± 2 °C). After that, the print was rinsed in deionized water and was left to dry in the oven at 50 °C for 30 min. The strips of filter paper used for the test were left to dry in free air. For assessment of the prints’ resistance to water, the printed test piece was brought into contact with filter papers previously saturated with distilled water and was placed under a 1 kg load for 24 h at ambient temperature (23 ± 2 °C). After the treatment, the laboratory prints were dried in an oven for 30 min at a temperature of 40 °C. The strips of filter paper used for the test were left to dry in free air. For ethanol stability assessment, the tests were made with two different concentrations of denatured ethanol (v/v = 96% and v/v = 43%, respectively) and the test was performed in a test tube, which was half-filled with solvent. The printed test piece with an area of approx. 6 cm^2^ was submerged in the solvent and was left in it for 5 min. The test temperature was 23 °C. After exposure, the laboratory prints were dried in an oven for 10 min at a temperature of 40 °C. Summarized details of test conditions for conducted chemical resistance test methods are presented in Table 3.

### 2.5. Assessment of Prints’ Resistance to UV Radiation

For the evaluation of UV stability of thermochromic prints, the samples were exposed to UV radiation in a Solarbox 1500e device (CO.FO.ME.GRA), with control of temperature and UV radiation. All the samples were exposed to filtered xenon light for a period of 6, 12, 18, and 24 h, at a Black Standard Thermometer temperature of 40 °C, at irradiation of 550 W/m^2^. The UV indoor filter was used to change the xenon spectral curve into the ultraviolet range. The used filter in that form allows the simulation of sun rays filtered through a windowpane, that is, it simulates the conditions of internal exposure. Conducted tests cover the determination of the relative lightfastness of printed matter under laboratory conditions in accordance with standard procedures described in ASTM 3424 and EOTA TR 010 [44,45].

### 2.6. Visual Evaluation

The test pieces were photographed on a copper plate coated with nickel, which is part of the electrostatic circulator on which the measurement was carried out at the selected temperature. Visual observation of the test pieces can be a first indicator of the degradation process, such as the changes in coloration.

### 2.7. Spectrophotometric Measurements

Evaluation of chemical and UV degradation of prints was done by spectrophotometric measurements. Spectrophotometric measurements were conducted on laboratory prints before and after their exposure to previously mentioned treatments. Temperature-dependent spectral reflectance of the samples was measured (spectral region 430–700 nm, 1 nm step) by fiber-based USB 2000 spectrometer (Ocean Optics, Orlando, Florida, USA) using a 30 mm wide integrating sphere (ISP-30-6-R) with (8°:di) measuring geometry and 6 mm sampling port diameter. OceanView software by Ocean Optics was used to calculate the CIELAB *L**, *a**, *b** values considering the D50 illuminant and 2° standard observer.

For the evaluation of chemical and UV degradation, the color difference between the prints before and after the conducted treatments was calculated using the formula CIEDE2000 [46]. Each printed test piece was measured three times at three different positions on the print area. Spectroscopic measurements were done at two fixed temperatures for each thermochromic print: one below its T_A_, and another above its T_A_. For that reason, prints made with OY-12 ink were measured at 8 °C and 20 °C, whereas prints made with PP-31 ink were measured at 20 °C and 45 °C, respectively. The samples were heated on the surface of a water block (EK Water Blocks; EKWB d.o.o., Ljubljana, Slovenia).

Moreover, ink transfer or bleeding from the printed test piece to the receptor surface (filter paper) was also evaluated by measuring color change CIEDE2000 on the filter paper that was in contact with the printed test piece [46]. 

## 3. Results and Discussion

### 3.1. Results of Prints’ Chemical Resistance

Assessment of the prints’ resistance to chemical agents was done based on any color changes of the prints and in the receptor (filter paper) surface or testing solvent that has been in the contact with the print [43]. Color changes CIEDE2000 of the test prints and staining of filter papers were evaluated by spectroscopic measurements. This was done by comparison of the treated test piece to an untreated test piece. 

Interpretation of the measured color-difference parameter ΔE from the CIEDE2000 formula was done in relation to subjective visual perception (Table 4). If the calculated CIEDE2000 results are below 1, then the average eye of the observer cannot perceive the difference between the two colors. A very small difference between colors can be perceived when the results are between 1 and 3, but it is tolerated, that is, a color difference up to 3 is acceptable [47]. Moreover, from the printing industry’s point of view, Delta E tolerances are acceptable for commercial printing if they are below 4.0, and for packaging printing the tolerated differences are up to 2.0 [48].

In case of bleeding of ink into the filter paper, the print is deemed to be bled when the stain on the filter paper has ΔE(00) > 1.4 [43]. It is important to determine bleeding (ink transfer) form the print into receptor surface because it can happen, for example, that the appearance of the print has not changed, but the filter paper shows staining. Thus, evidence of bleeding is one of the indicators that the print is not resistant to chemical agents.

Figure 3 shows the results of measured CIEDE2000 values for prints obtained with PP-31 ink, for both paper substrates, after treatments with chemical agents. Figure 3a shows the results when the TC ink was below its activation temperature (20 °C), meaning that both colorants are in a colored state, whereas Figure 3b shows results obtained above the ink’s activation temperature (45 °C) when discoloration of the thermochromic colorant occurred. Results of measured ink bleeding from the printed test pieces to the filter paper expressed in CIEDE2000 values are also included in Figure 3 and presented as dots.

When the TC ink is below its activation temperature (TC colorant is colored), the color difference results are noticeably smaller than in the case when thermochromic colorant becomes transparent/translucent. When prints are measured at 45 °C, it can be noticed that for most liquid agent the prints were exposed to, color difference values go above the value of 3, which is an unacceptable color change that can be perceived visually (appreciable perception of color difference). In the case when thermochromic colorant is in its colored state, measured color differences are acceptable, except for the result measured on a CHR test piece that was submerged in 96% ethanol. Moreover, when comparing two printing substrates, prints show better stability when printed on embossed NPF paper than on ultra-smooth CHR paper. In Figure 4, the stains on filter papers are visible indicating bleeding of the colorants from prints exposed to water and citric acid. The CIEDE2000 values obtained for colorants bleeding determination to the filter papers are pretty much similar regardless of the temperature at which they were measured, which may indicate that mostly non-thermochromic colorants were transferred to the receptor surface, or that thermochromic colorants were not transferred to a significant level (which coincides with measured smaller color differences when both colorants were activated, as presented in Figure 3). It should also be noted that the colorants bled more when exposed to citric acid than to water, which can also be seen in Figure 4. Moreover, a higher degree of bleeding was noticed from prints obtained on CHR papers into filter paper saturated with water, whereas, in the case of prints’ exposure to citric acid, slightly higher bleeding occurred from prints obtained on NPF papers. In the case of prints obtained on CHR papers, migration from the printed side of the test piece to the back side was noticed, that is, filter paper that was put below the print was also stained with ink. Bleeding was visually observed on all four filter papers that were put below and above the CHR test pieces during the experiment, while this was not observed for test pieces printed on NPF paper. 

Figure 5 shows bleeding of the colorants of both tested TC inks, OY-12, and PP-31 into ethanol solutions (v/v = 43% and v/v = 96%). The first test tube on the left, contains only the neat solvent, whereas the others contain also the colorants that bled into solutions after the treatment of the test pieces (bleeding of the colorants can be observed, especially in the case of the test pieces printed with PP-31 ink). The results show that more severe bleeding occurred when test pieces were treated with 96% ethanol solution and that ink transfer into the solvent is greater from prints obtained on CHR papers (which corresponds with the color difference results presented in Figure 3).

Figure 6 shows the results of measured CIEDE2000 values for prints obtained with OY-12 TC ink, for both paper substrates, after treatments with chemical agents. Figure 6a shows results when the ink was below its activation temperature (8 °C), meaning that both colorants are in a colored state, whereas Figure 6b shows results obtained above the ink’s activation temperature (20 °C) when discoloration of the thermochromic colorant occurred. Results of measured ink bleeding from the printed test pieces to the filter paper expressed in CIEDE2000 values are also included in the figures and presented as dots.

By observation of the results presented in Figure 6, it can be interesting to see that the color difference results are significantly smaller when thermochromic colorant is inactive, that is, when measured above its activation temperature. These are the complete opposite results when compared to results obtained with PP-31 ink. When prints are measured at 8 °C, when the thermochromic colorant is in its colored state, it can be noticed that liquid agents for all examined test pieces caused discoloration of prints as the color difference values go above the value of 5 and even 6, which is an unacceptable color change. When comparing two printing substrates, prints, in most cases, show better stability when they are printed on embossed NPF paper, than on ultra-smooth CHR paper. CIEDE2000 results that refer to the bleeding of the colorants point out that ink has bled when prints were exposed to water and citric acid (ΔE(00) > 1.4); however, when stains on filter papers were measured above ink’s activation temperature, measured CIEDE2000 results were, in most cases, smaller and close to the threshold value (ΔE(00) = 1.4). Also, when compared to the PP-31 ink, OY-12 ink showed a lesser degree of bleeding. Due to the weak contrast of the yellow color on the white filter papers, it was hard to visually notice the bleeding of colorants, which is why the image of bleeding is not presented as in the case of PP-31 ink. Bleeding of the colorants into the ethanol solutions was observed visually (Figure 5) but also to a much lesser extent when compared to the PP-31 ink. Also, a higher concentration of the solvent (96% ethanol) caused the ink to bleed more.

Taking into account the results presented earlier, it can be concluded that CIDE2000 values obtained at temperatures lower than the activation temperature result from both colors of TC microcapsules and conventional pigment, whereas CIDE2000 values obtained at a temperature higher than the activation temperature result mostly from the color degradation of the classic pigment. Also, the stability of TC prints towards different chemicals decreases with decreasing solvent polarity. It is known that water is a polar solvent, while with organic molecules, this influence is reduced due to the hydrocarbon chain. Considering the fact that citric acid is a weak organic acid (carboxylic acid) and that due to the presence of three carboxyl groups and one hydroxyl group, it has a higher polarity than ethanol (one hydroxyl group). The binder present in the used inks and the conventional pigment create an interaction with the used solvents, and in this way are transferred to another medium (filter paper or solvent). Water and citric acid interact mostly with hydrogen bonding, whereas ethanol reacts by polar interactions (Table 3) [32,50]. In 43% ethanol solutions hydrogen interactions predominate, with somewhat lower dispersive interactions. In 96% EtOH solution, the polar interactions are dominant. With that in mind, we can conclude that the used printing ink interacts mostly by high polar interactions as evident from the high color difference. It is also important to emphasize that in the tests in which bleeding was observed, it cannot be asserted with certainty whether it is only a matter of bleeding of the pigment, but it should be emphasized that in this case bleeding of the resin, which is the carrier of the pigment inside the printing ink, is also possible. Therefore, this is one limiting factor of this study that will be considered more in the future. In addition to the influence of chemicals, the influence of the printing substrate on the chemical stability of prints is also visible. A printing substrate with a rougher surface (NPF) shows better stability. This can be attributed to the adhesive properties of the ink, which arise precisely because of the roughness of the surface. Besides physical and chemical adhesion (formation of ionic, covalent, hydrogen bonds) occurring on the paper-printing ink interface, mechanical adhesion occurs by the penetration of printing ink into surface irregularities (pores) of the printing substrate [51]. Thus, the rougher the surface, the better the stability of TC prints. 

### 3.2. Results of Prints’ UV Stability

Evaluation of UV degradation of the prints was done by spectroscopic measurements and expressed in color difference values CIEDE2000. This was done by comparing untreated samples with treated samples after 6, 12, 18, and 24 hours of exposure to UV radiation. Exposure to UV radiation diminishes the dynamic of the color of TC prints as is clearly visible from Figure 7 and Figure 8.

Color difference results on both PP-31 and OY-12 prints are significantly smaller when thermochromic colorant is inactive that is, when measured above its activation temperature. Such results could probably mean that UV radiation caused greater degradation of TC colorants, although it is evident from the results that conventional colorants are also strongly affected by UV radiation. When comparing NPF and CHR printing substrates, it can be seen that prints show better stability when they are printed on embossed NPF paper.

Thermochromic printing inks are colored complex mixtures, consisting of colorants (conventional pigments and thermochromic microcapsules), binder (resins), and different additives. Their composition and physical property mostly differ due to the intended printing process. Moreover, thermochromic (TC) microcapsules, which are one of the main components, consist of dye–developer–solvent systems encapsulated inside the polymeric shell. Polymeric shell of TC microcapsule is mostly made of melamine resin, but in this case, we can not make claims about its origin. According to previous research, the binder of used UV-curing TC printing ink is based on polyurethane acrylate (PUA) [52]. In addition, the TC microcapsules in the print, along with conventional pigments, are covered with ink’s binder [52]. In the case of TC composite, the intensity of color contrast when mixing TC composit is affected by the intermolecular reactions between color former, color developer, and organic solvent [39,40]. Taking all of that into account, when the UV radiation comes over the surface of TC print, different chemical reactions are involved in this system. UV radiation causes damage to materials through a complex photochemical reactions, where polymeric materials first absorb the energy which is afterwards utilized for the breakage of the molecular bonds. The presence of chromophore groups causes faster materials degradation due to its high capacity of UV absorption [53]. According to Groeneveld et al., photodegradation of organic colorants, which are one component of TC microcapsules, is a dynamic process influenced by a number of different internal (physical state of the dye- absorption spectrum and polarity) and external (spectral distribution and intensity of the light source, presence or absence of oxygen, temperature, humidity, pH, the type of solvent or substrate, and concentration of dye) parameters [54]. When dye adsorbs light, different protective pathways occur and often are followed to lose the excess energy by emitting a photon, through non-radiative relaxation or by molecular reactions (photoreactions). In the case of photoreactions, photodegradations occur due to chemically unstable dye molecules in their photoexcited form, which in the end results in their decomposition through dissociation, intramolecular rearrangement reactions, or redox processes. Due to photoexcited chemically unstable dye molecules with the presence of other reactive substances in the system, both initiated from the singlet or triplet excited state (^1^D*, ^3^D*) of the dye molecules [54]. In indirect photochemical reactions, the compounds with high absorption coefficients and low activation energies (photosensitizers), when excited can react directly with the neighboring dye or pigment or they can react with molecular oxygen (^3^O_2_) to create singlet oxygen (^1^O_2_), which then becomes the reactive species [54]. Friškovec et al. showed that during photooxidation, TC microcapsules shells in the thermochromic printing ink can be destroyed resulting in an irreversible loss of its functional properties because the polymer shell does not protect the TC composite from the environment [33]. Taking into account the ink binder, the photodegradation of PUA probably involves the Norrish type I photocleavage of the excited carbonyl groups in acrylate and carbamate moieties [55,56,57,58]. In polyurethane-based materials, the urethane linkages (C-NH) are most susceptible to photodegradation which, in the end, causes the surface erosion of the top layers and the formation of oxidation products consisting of hydroxyl type compounds (alcohols, hydroperoxides) and carbonyl compounds [55]. In addition, to C-NH groups, the methylene (CH) groups are damaged during UV exposure of PUA [55]. The addition of UV absorbers and the piperidine hindered amines (HALS) can be a promising way to increase the UV stability of PUA [56]. Moreover, the photostability of color formers of TC composite can be acieved by the addition of the amphoteric counter-ion (benzophenone or benzotriazole type UV absorbers, naphthalene derivatives of benzotriazole type UV absorbers, and zinc and nickel 2,4-dihydroxybenzophenone-3-carboxylates) according to the literature [39,40,41].

Keeping that in mind, if the print is exposed to UV radiation, first the binder degrades and results in the formation of oxidative species, which in the end react with the TC microcapsules shell and cause the color degradation. The UV stability of TC printing inks thus is affected by various factors such as the chemical composition of the binder and the TC microcapsules, interactions between the printing ink’s binder and microcapsules, and the binder drying mechanism [37]. With that in mind, it can be concluded that TC printing ink is a complex mixture in which during photooxidation, different photochemical reactions occur, causing the formation of oxidation products and fast oxidation of a whole system—the ink binder, TC microcapsules polymer shell, dye-developer-solvent system, and conventional pigment. Due to a complex structure of TC printing inks, the photodegradation should be considered on different levels such as the increased photostability of color formers and color developers in TC composites, TC microcapsules shells, and printing ink binders.

TC inks are considered to work properly if the total color contrast (TCC) between the two states, below and above the activation temperatures, is clearly recognized. TCC was measured as the CIEDE2000 difference between samples at 8 °C and 20 °C for OY-12 ink. For PP-31 ink, TCC was measured as the CIEDE2000 difference between samples at 20° and 45 °C. The higher the TCC value, the more pronounced the thermochromic effect will be.

From Figure 9, it can be seen that the TCC is significantly higher on the embossed NPF paper. Although on untreated samples TCC of PP-31 ink printed on ultra-smooth CHR paper is higher than on NPF paper, already after 6 h of UV radiation its TCC contrast is much lower compared to PP-31 ink printed on NPF paper. In comparison, after 18 hours, TCC on CHR papers is 1.15 CIEDE2000 units for PP-31 and 1.48 units for OY-12. On NPF papers after 18 hours of UV radiation, the TCC is 3.61 CIEDE2000 units for PP-31 and 7.07 units for OY-12. The reason is probably that the NPF paper surface is more structured and the ink penetrates more into the paper structure irregularities compared to the ultra-smooth CHR paper as explained earlier. Photodegradation is strongly influenced by the amount of ink present on a substrate surface which can lead to aggregation of ink concentration. The size of the aggregates is proportional to ink concentration. If the agregates are greater, the lightfastness improves due to the smaller relative surface area accessible to environmental factors responsibe for photofading [54]. Therefore, it can be concluded that if the prints will not be exposed to UV radiation, that is, they will be intended for use in closed spaces, the TCC effect will be more pronounced on smooth substrates, as can be clearly seen in Figure 9. Degradation of prints due to UV radiation on both substrates is quite large; however, due to their structure, embossed papers are a slightly better choice when printing with TC inks.

#### Visual Evaluation of Prints

The result of the visual evaluation of the samples photographed on the plate on which the samples were cooled and heated for spectrometric measurement can be seen in Figure 10, Figure 11, Figure 12 and Figure 13. It is evident that the prints, which were exposed to UV radiation, significantly differ from untreated samples in color. Significant changes of color can be seen even after 6 h of exposure to UV radiation, whereas after 18 h of UV radiation, the loss of dynamic color change can be noticed. The degradation of both TC inks is more significantly visible on the smooth CHR substrate On prints obtained with both TC inks (OY-12 and PP-31) the degradation is smaller above the activation temperature when the TC colorants are inactive. 

## 4. Conclusions

The main task of thermochromic inks is to influence the experience of the message or design printed with them through their dynamic color change. Their role can be informative or just creative. By analyzing some factors affecting the degradation of TC prints, that is, the dynamics of color change, an insight into the weaknesses of this complex mixture was provided. 

All tested TC prints showed low resistance to liquid chemical agents as color difference values were unacceptable in all cases and bleeding of the colorants was detected as well. It was noticed that the stability of TC prints towards different chemicals decreases with decreasing solvent polarity, and the poorest stability was observed in the case of prints’ exposure to a 96% ethanol solution. Prints obtained with body-heat activated TC ink (PP-31) showed higher color difference results when measured below ink’s T_A_ (when both colorants were in their colored state), whereas in the case of prints obtained with cold-activated TC ink (OY-12), higher color differences were detected when prints were measured above ink’s T_A_ (when TC colorant was inactive). Higher bleeding was detected both visually and by spectroscopic measurements from body-heat activated TC PP-31 ink, than from cold-activated TC OY-12 ink.

This research also provided insight into the relationship between TC inks and the substrate on which they are printed. Both inks showed better stability when printed on embossed NPF paper than on ultra-smooth CHR paper. Moreover, in the case of ultra-smooth CHR paper, bleeding of TC PP-31 ink’s colorants was visually observed on all filter papers that were put below and above the prints during the experiment, meaning that the migration from the printed side of the test piece to the back side was also noticed.

Due to the sensitivity of TC inks to adverse environmental conditions, the goal of this research was to determine the resistance of TC prints to the influence of liquid chemical agents and UV radiation. Based on the results obtained after UV radiation, its influence in terms of color degradation, that is, reduced dynamics of color change is visible on both tested paper substrates. However, more significant degradation, reflected as a reduced total contrast, was observed on the ultra-smooth paper substrate. The reason is probably that embossed paper, due to its structure, enables better ink penetration into the irregularities of the paper structure compared to ultra-smooth paper. Also, in prints that were not exposed, it was found that the total color contrast is more noticeable on ultra-smooth paper than on embossed paper.

## Figures and Tables

**Figure 1 polymers-15-01208-f001:**
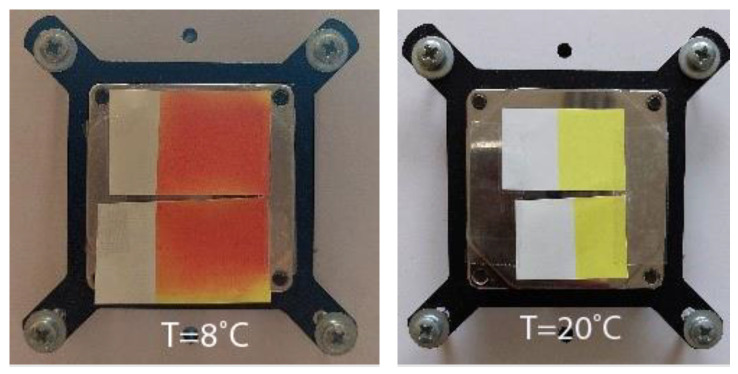
Visual presentation of thermochromic prints obtained on different paper substrates of cold activated OY-12 ink under (8 °C) and above (20 °C) ink’s T_A_.

**Figure 2 polymers-15-01208-f002:**
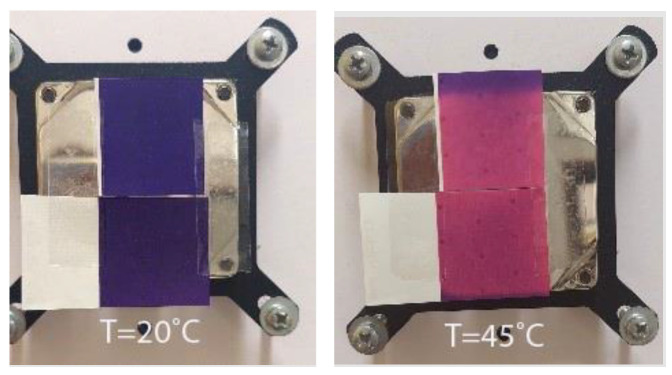
Visual presentation of thermochromic prints obtained on different paper substrates of body-heat activated PP-31 ink under (20 °C) and above (45 °C) ink’s T_A_.

**Figure 3 polymers-15-01208-f003:**
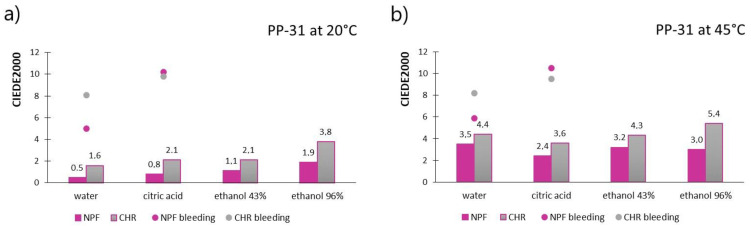
CIEDE2000 measured on PP-31 prints below ink’s T_A_ (**a**) and CIEDE2000 measured on PP-31 prints above ink’s T_A_ (**b**).

**Figure 4 polymers-15-01208-f004:**
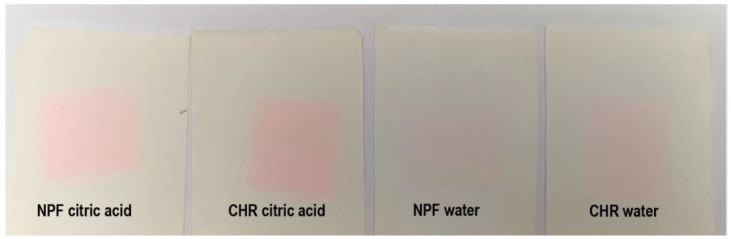
Bleeding of colorants into filter papers from test pieces printed with PP-31 TC ink.

**Figure 5 polymers-15-01208-f005:**
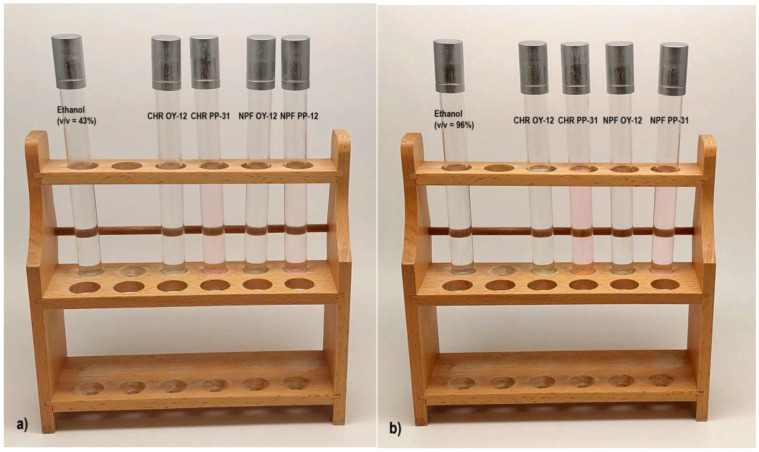
Bleeding of colorants from prints obtained with OY-12 and PP-31 TC inks into solvents: (**a**) 43% ethanol solution and (**b**) 96% ethanol solution.

**Figure 6 polymers-15-01208-f006:**
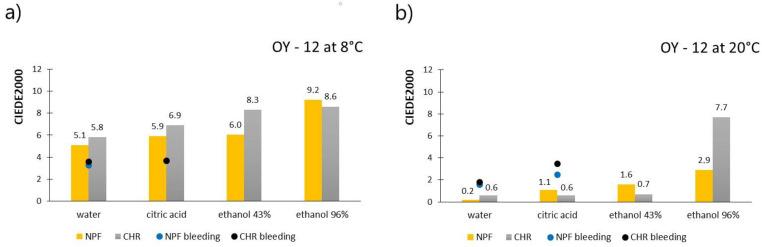
CIEDE2000 measured on OY-12 prints below ink’s T_A_ (**a**) and CIEDE2000 measured on OY-12 prints above ink’s T_A_ (**b**).

**Figure 7 polymers-15-01208-f007:**
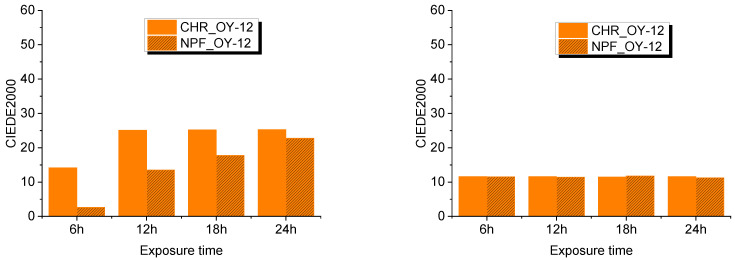
CIEDE2000 difference measured at 8 °C (**left**) and at 20 °C (**right**) on both test substrates printed with OY-12.

**Figure 8 polymers-15-01208-f008:**
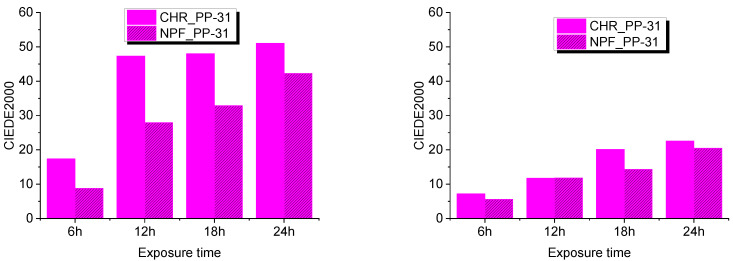
CIEDE2000 difference measured at 20 °C (**left**) and at 45 °C (**right**) on both test substrates printed with PP-31.

**Figure 9 polymers-15-01208-f009:**
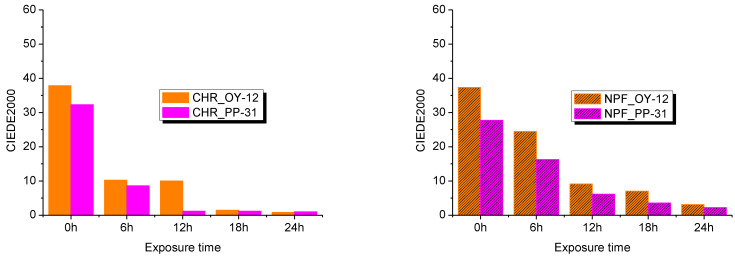
The total color contrast (TCC) between the two states for the OY-12 and PP-31 prints before and after 6, 12, 18, and 24 h exposure to UV radiation.

**Figure 10 polymers-15-01208-f010:**
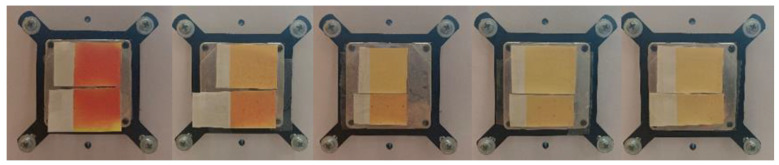
Visual presentation of thermochromic prints of cold activated TC ink OY-12 at 8 °C, before and after exposure of samples to UV radiation unexposed and exposed (6, 12, 18, and 24 h) (above CHR sample, below NMF sample).

**Figure 11 polymers-15-01208-f011:**
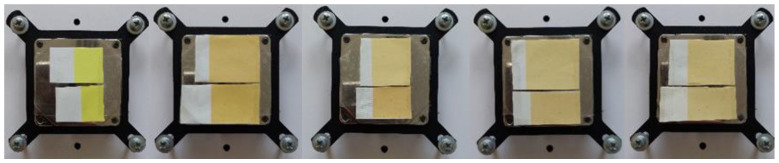
Visual presentation of thermochromic prints of cold activated TC ink OY-12 at 20 °C, before and after exposure of samples to UV radiation (6, 12, 18, and 24 h) (above CHR sample, below NMF sample).

**Figure 12 polymers-15-01208-f012:**
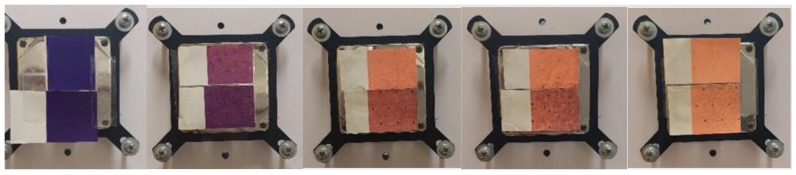
Visual presentation of thermochromic print of heat activated TC ink PP-31 at 20 °C, before and after exposure of samples to UV radiation (6, 12, 18, and 24 h) (above CHR sample, below NMF sample).

**Figure 13 polymers-15-01208-f013:**
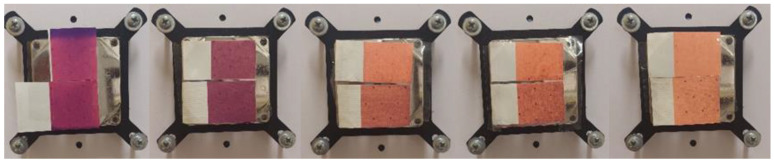
Visual presentation of thermochromic print of heat activated TC ink PP-31 at 45 °C, before and after exposure of samples to UV radiation (6, 12, 18, and 24 h) (above CHR sample, below NMF sample).

**Table 1 polymers-15-01208-t001:** Properties of used printing substrates.

Paper Substrate	Abbreviation	Property Unit Method	Basis Weight (g/m^2^) ISO 536	Caliper (µm) ISO 534	ISO Brightness (%) ISO 2470, R457	Bekk Smoothness (s) ISO 5627
Niklakett Premium Fashion	NPF		75	74	91	74
Chromolux 700	CHR		80	84	89.3	300

**Table 2 polymers-15-01208-t002:** Characteristics of used thermochromic UV screen printing inks.

Property	Value
Viscosity at 25 °C	65–110 poise
Density (Approx.)	8.0 lb./gal
Appearance	Viscous Liquid
Percent Solids (Approx.)	99%
Percent Volatiles (Approx.)	<1.5%

**Table 3 polymers-15-01208-t003:** Test conditions for used liquid agents [43].

Test Agent	Receptor Surface	Temperature/°C	Test Duration	Contact Condition
Water (distilled)	filter paper	23 ± 2	24 h	1 kg on 54 cm^2^
Citric acid (w/v = 5%)	filter paper	23 ± 2	1 h	1 kg on 54 cm^2^
Ethanol (v/v = 96%)	test tube	23 ± 2	5 min	liquid
Ethanol (v/v = 43%)	test tube	23 ± 2	5 min	liquid

**Table 4 polymers-15-01208-t004:** Subjective assessment metric based on CIEDE2000 Color difference [47,49].

Color Difference	Perception of Color Difference	Tolerance
<0.2	not visible	Acceptable for the printing industry
<0.5	negligible	
0.2–1.0	noticeable	
1.0–3.0	visible, but small	
3.0–6.0	clearly visible, obvious	Unacceptable for the printing industry
6.0–12.0	extremely large	
>12.0	unacceptable	

## Data Availability

Not applicable.

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
