# Peer review of "Assessment of Thermochromic Packaging Prints’ Resistance to UV Radiation and Various Chemical Agents"

_polymers, 2023, doi:10.3390/polym15051208_

Round 1

Reviewer 1 Report

Comments

   The study relates to the chemical resistance and UV resistance of TCs, which are commonly used, and observed color changes caused by two environmental factors. Systematic observations have been made, but they are about predictable phenomena that already exist.  It is necessary to describe the academic meaning of this phenomenon in the paper.  In addition, it is necessary to directly check (including formula) how each TC has changed chemically.

1. In the case of the paper used in the study, it is used as a label in the product. In this case, it is expected that there will be no environment in direct contact with ethanol or citric acid excluding water for a long time. Is there a common case of long-term exposure to chemical agents as in the conditions of this study?

2. Many thermochromic inks have been commercialized for sale (at least 10). Despite the data studied on errors that appear after printing, such as coloring and color bleeding, there are only two test cases.

3. It is necessary to describe the meaning of observing the color change of bleeding.

4. The change according to UV radiation is not very large, and since information is conveyed with a clear color change, slight discoloration or damage does not seem to be of great significance.

5. In order to examine the effectiveness of the actual application, it is expected that long-term exposure compared to the experimental conditions of this study is required.  It is necessary to confirm whether the current experimental conditions are simulated by accelerating the environment in the actual situation.

6. The results obtained through spectroscopic analysis are also significant, but since the human eye will see the package, it seems necessary to compare the results obtained through the actual photograph and the spectroscopic colorimeter.

7. Line 246 specifies 'Unacceptable Color Change', but what are the quantitative criteria for distinguishing Acceptable and Unacceptable?

8. Some errors need to be corrected.

- 2.3 and 2.2 have the same title

-It is suggested to display upper case and lower case letters separately in the "Keywords" part.

- The left graph of Figure 8 does not overlap the results and legends.

- Requires confirmation of table of contents number (Nothing related to “ 4. ”)

Reviewer 2 Report

This is a useful and well conducted study but  in the current form provides little information to the reader, since the chemical structures of the coloured compounds are only described in a limited and general way. In order to understand the degradation process we need to have details of the structures involved.  If radical formation is involved here, for example via singlet oxygen formation, then it would clearly help to know what molecular features are likely to be damaged in this process.  Certainly presenting UV vis absorption spectra would help; as would a figure showing the  wavelength intensity profile of the UV light source.  

I understand some proprietary information is involved in these materials but in order to fully understand and compare them some information on the presence of stabilisers and relative concentrations of such would be useful; or indeed whether there are any such materials which might improve the stability ?

Reviewer 3 Report

1.    The abbreviation BST in Page No. 5, Line No.184 must have been defined earlier to its use.

2.    How the test duration was fixed for distilled water, citric acid and ethanol? Is it according to standards?

3.    Ocean View software calculated values for L*, a*, b* can be presented in in the manuscript to improve the understanding of readers.

Author Response

We would like to thank reviewer for pointing out problems in the manuscript. We hope that we have succeeded in addressing and explaining the questionable parts of the manuscript.

Our response to the reviewer is attached in the document below. 

Best regards,

The Authors
